# Bridging the knowledge-practice gap: A cross-sectional survey assessing physician knowledge, attitude and practice toward complementary and alternative medicine

Sarah Salih[1]*, Tif A. Jawahi[2], Rafif H. Al Salem[2], Shahad A. Alhazmi[2], Atheer A. Buayti[2], Arwa H. Alammari[2], Hadeel M. Mashi[2], Layla A. Dobea[2], Mohammed A. Muaddi[1]

1 Family and Community Medicine Department, Faculty of Medicine, Jazan University, Jazan City, Saudi Arabia, 2 Faculty of Medicine, Jazan University, Jazan City, Saudi Arabia

* ssalih@jazanu.edu.sa

## Abstract

Complementary and alternative medicine (CAM) is widely used in multiple countries, including Saudi Arabia, yet its integration into mainstream healthcare requires evidence-based guidance. However, little is known about the current level of awareness attitudes and practice of physicians regarding CAM in Jazan region, which is a predominantly rural region in the southwestern part of Saudi Arabia. This region is known for its diverse residents with unique cultural and healing practices. The study aimed to assess physicians' awareness, attitudes, and practices regarding CAM in the Jazan region. We conducted a cross-sectional survey among physicians in five governorates of Jazan using convenience sampling. The structured questionnaire explored CAM-related awareness, attitudes, and practices. Data were analyzed using descriptive and inferential statistics to identify associated factors. Of 159 responding physicians (58.5% male and 41.5% females), 81.1% were aware of CAM, but only 7.5% had received formal pre-service training. Attitudes towards CAM were generally positive, with a median attitude score of 23 (out of a total of 30), particularly towards the need for more research and education on CAM. However, despite the generally positive attitudes toward CAM, only 25.8% of physicians reported plans to integrate CAM into their practice, and 37.7% had previously recommended CAM therapies to their patients. Factors significantly associated with awareness and practice included age, gender, work experience, and healthcare setting. In conclusion, despite high awareness and positive attitudes towards CAM among Jazan physicians, there is a significant gap in formal training and limited integration into practice, a matter which underscores the need for further exploration of the factors influencing this discrepancy.

**Data availability statement:** The datasets used and/or analyzed during the current study are available as supplementary information under the name S1_dataset. The survey instrument used is available as supplementary file S2_Instrument.

**Competing interests:** I have read the journal's policy and the authors of this manuscript declare that they have no competing interests.

**Abbreviations:** CAM, Complementary and alternative medicine; KAP, Knowledge, attitudes and practice.

## Introduction

Complementary and alternative medicine (CAM) has recently garnered significant attention as a potential adjunct or alternative to medical treatment modalities. CAM is defined as a diverse group of healthcare practices which are not considered a part of conventional medical therapies [1]. These practices include a broad range of modalities, such as acupuncture, herbal medicine, chiropractic care, and homeopathy. Globally, CAM is used on a wide scale, with estimates indicating that 36% of adults in the United States have utilized one or more forms of CAM, and the prevalence is higher in some countries [2]. Early evidence from European countries showed that 20%-50% of individuals have used CAM modalities, with highest estimates in Germany and France (46% and 49%, respectively) [3]. In Australia, approximately 68.9% of the population have reported using CAM, particularly yoga, meditation and chiropractic care [4]. Studies published in Asian countries revealed varied utilization rates of CAM practices, such as traditional Chinese medicine, Ayurveda, and acupuncture, with highest rates reported in Singapore (76.0%) [5], South Korea (75%) [6].

Actually, CAM therapies have been increasingly supported by scientific evidence worldwide demonstrating the effectiveness of these therapies in reducing pain, managing chronic illnesses and enhancing overall well-being [7]. In Arab and Islamic regions, CAM therapies represent an integral part of religious and cultural traditions. Black seed and honey have shown promising outcomes due to their antimicrobial, anti-inflammatory and immune-boosting properties [7]. Cupping therapy have also demonstrated effective results in alleviating musculoskeletal pain and improving the quality of life of patients with chronic pain [8]. Ruqyah (spiritual healing) and honey had also important roles in improving mental health and reducing stress, and herbal medicine remedies are increasingly utilized to reduce symptoms and improving the quality of life of patients with mental health conditions [7,8].

In Saudi Arabia, the use of CAM modalities is deeply rooted in local traditions due to cultural, social and religious aspects. National studies showed that a majority of individuals (70%) engaged in CAM practices, such as prophetic medicine and natural remedies [9]. CAM regulation in Saudi Arabia is overseen by the National Center for Complementary and Alternative Medicine (NCCAM), which has been established under the Ministry of health in 2009 [10]. NCCAM is responsible for regulating CAM practices, licensing clinics and practitioners and enhancing public awareness. The 2019 NCCAM regulation represented a significant advancement in the governance of CAM practices, where only licensed practitioners are allowed to offer CAM services after passing standardized exams [10,11]. NCCAM has the authority to add or suspend practices beyond five approved modalities (cupping therapy, naturopathy, acupuncture osteopathy, and chiropractic), a matter which reflects the country's commitment to ensure the efficacy and safety of CAM [10].

Actually, the increasing trend of CAM use necessitates physicians and other healthcare providers to possess adequate knowledge and positive attitudes towards CAM to facilitate effective communication with patients and ensure the use of CAM safely. However, studies regarding the knowledge, attitude and practice of CAM among physicians in different Saudi regions showed a wide range of findings. For

example, residents in the Tabuk region indicated that CAM could be an acceptable adjunct to conventional medical therapies, with a considerable proportion recommending CAM use [12]. Additionally, a study from Madinah showed that physicians highlighted the importance of CAM; yet, the majority of them lacked adequate knowledge to effectively counsel patients [13].

These findings were also applicable in Jazan, where a recent study showed that a small proportion of primary healthcare physicians (12.8%) had attended CAM-related lectures or training, and almost one-third of them (37.6%) had ever individually used CAM [14]. Interestingly, although most healthcare professionals in Riyadh reported knowledge about CAM, many of them had acquired their knowledge through informal, non-medical sources [15], which might influence their evidence-based recommendations and communications with patients. While studies have been conducted in various regions of Saudi Arabia regarding CAM, the situation in Jazan remains largely unexplored. The Jazan region has unique geographic, cultural and healthcare landscape, since the population is this area is predominantly rural, with strong adherence to traditional and cultural practices [16,17]. Additionally, there is evidence that the access to advanced healthcare services is limited in some local areas [18], which makes the choice of CAM a popular approach. Previous research in other areas has revealed varying CAM knowledge and usage levels among healthcare professionals, with many acquiring information through informal channels [12,15]. However, the specific landscape of CAM awareness, attitudes, and practices among physicians in Jazan's healthcare institutions is poorly understood [14]. This knowledge gap is significant, as it impedes the development of targeted strategies to enhance CAM integration and patient care in the region.

To address this critical issue, the current study aims to assess the knowledge, attitudes, and practices (KAP) of physicians regarding CAM in five governorates in Jazan. Additionally, we seek to identify factors that may influence KAP, which could inform future educational initiatives and policy development. By gaining insights into the current state of CAM engagement among Jazan's medical professionals, this study will provide a foundation for evidence-based decisions to improve healthcare delivery and potentially bridge any existing gaps between conventional and complementary medicine in the region.

## Materials and methods S

### Study design and population

An observational, cross-sectional study was conducted among physicians working in primary, secondary, and tertiary hospitals in five governorates in Jazan region (Jazan, Abu-Arish, Sabya, Damad, Baish). Eligible participants included Saudi and non-Saudi physicians, with no gender or age limitations. Visiting physicians or those who are not English speakers were excluded. The exclusion of visiting physicians was adopted due to their temporary service status, which might not actually reflect their typical practice and attitudes in the Jazan region compared to those with permanent services. Additionally, non-English speaking physicians were excluded to ensure that respondents have fully understand questions without misinterpretation. This is because the English language being the primary medium of instruction in medical schools. This way, including participants with a dominant English language would improve the validity and reliability of responses [19].

### Ethical considerations

This study was conducted in strict adherence to ethical research standards, with approval obtained from the Jazan Health Ethics Committee (approval number: 22134; Date: 20/12/2022) before its commencement. All participants were thoroughly informed about the study's purpose and methodology, and their informed consent was secured before participation. All responses were anonymised to protect participants' privacy and maintain confidentiality, with no personal identifiers collected or stored. Participants were assured that their responses would remain confidential and only be analyzed and reported in aggregate form. The voluntary nature of participation was emphasized, and individuals were informed of their

right to withdraw from the study at any time without any negative consequences. These measures ensured the protection of participants' rights and welfare throughout the research process while maintaining the integrity and validity of the collected data.

## Sampling technique and sample size

The Jazan region contains 21 hospitals employing a total of 2104 physicians, based on data from the most recent report from the Ministry of Health, Saudi Arabia, in 2021 [20]. Based on a knowledge level of 87.5% among physicians, as retrieved from a previous publication [15], the sample size was estimated to be 159 physicians. Physicians were recruited from the five governorates using a convenience sampling method. Each region was treated as a stratum, with the number of participants proportionally allocated based on the total number of physicians working in healthcare facilities in each region. The convenience sampling method was used to allow efficient recruitment and inclusion of diverse healthcare settings. Efforts were made to reduce the potential selection bias (physicians who were more available or willing to participate may differ in their awareness) and enhance the representativeness of participants. This was done by recruiting participants from a variety of governorates and healthcare settings. The sample size was calculated using a 95% confidence interval and a margin of error of 5% using the following formulas[21]:

$$x = Z^2 \times r \times (100 - r)$$

$$n = \frac{N \times x}{(N-1)E^2 + x}$$

$$E = \sqrt{\frac{(N-n)\, x}{n \times (N-1)}}$$

Where N is the population size (2104 physicians), r is the response distribution (87.5%), Z is the critical value for a confidence level of 95% (1.96).

## Data collection tools

Data were collected from 24/01/2023 till 06/04/2023 by seven data collectors trained to deal with participants and expected difficulties in data collection. Data were collected based on a previously validated questionnaire [22]. Details about the used questionnaire are provided in the supplementary file (S2_ Instrument in S2 File). The survey instrument underwent a rigorous validation process specifically adapted for the Jazan context. Initially, the questionnaire was developed in English based on comprehensive literature review. Content validity was established through review by a panel of local experts (n = 5) including CAM specialists, research methodologists, and practicing physicians from Jazan region who assessed item relevance, clarity, and cultural appropriateness. The Content Validity Index (CVI) was calculated, with items achieving a minimum CVI of 0.80 being retained.

Face validity was established through pilot testing with 10 physicians from different healthcare facilities in Jazan who were not included in the final sample. The pilot testing assessed comprehension, cultural appropriateness, and time required for completion. Based on pilot feedback, minor modifications were made to improve clarity and cultural relevance of certain items. The internal consistency reliability was assessed using Cronbach's alpha, which was 0.81 The questionnaire consisted of five sections: 1) demographic characteristics, including physicians' age, gender, nationality, marital status, job level, years of work experience, current health facility, and specialty; 2) Awareness of

CAM, including questions on whether the physicians had heard about CAM, the sources of their information, the types of CAM treatment options they were aware of, and whether they were aware of the harmful effects of CAM; 3) CAM Training, including whether participants had received pre-service training in CAM and, if so, the source of that training; 4) Attitudes towards CAM, including six questions which explored their views on combining complementary and modern medicine, increasing patient satisfaction, the need for medical education in CAM, incorporating CAM into the medical curriculum, conducting research on CAM efficacy and safety, and the benefits of wellness centers combining both types of medicine; 5) Practice and recommendations for using CAM, including questions about participants' current practices and recommendations related to CAM.

**Attitude score calculation.** Participants' attitudes toward complementary and alternative medicine (CAM) were assessed using six questions, each rated on a five-point Likert scale ranging from "strongly disagree" (scored as 1) to "strongly agree" (scored as 5). An overall attitude score was calculated by summing the scores of all six items, resulting in a total score ranging from 6 to 30. A score higher than the median indicated a positive attitude toward CAM.

**Statistical Analysis.** Data were analyzed using RStudio software ((R Foundation for Statistical Computing, Vienna, Austria, version 4.3.1). Descriptive statistics were used to summarize participants' sociodemographic characteristics, awareness, attitudes, and practices regarding complementary and alternative medicine (CAM). Categorical variables were presented as frequencies and percentages. To assess the association between participants' awareness, attitudes, and practices toward CAM and their sociodemographic characteristics, Pearson's Chi-squared test and Fisher's exact test were applied where appropriate. The significance level was set at $p < 0.05$.

## Results

### Sociodemographic and occupational characteristics

Initially, we collected data from 166 physicians, of whom 7 refused to participate.

As shown in Table 1, the study population predominantly consisted of young physicians, with nearly two-thirds under 35 years old. There was a moderate male predominance (58.5%) whereas 41.5% of the sample were females. Most participants were Saudi nationals (71.9%). Professionally, residents constituted the largest group (44.0%), and most physicians (67.9%) had less than 10 years of work experience. Half of the participants worked in primary healthcare settings, with primary care and general practice being the predominant specialty (51.9%).

### Characteristics of participants' awareness and training of CAM

The majority of physicians (81.1%) reported awareness of complementary medicine, primarily gaining their knowledge through media sources (70.3%). Among CAM modalities, Hijama (cupping) and massage were the most widely recognized (73.4% and 72.7% respectively). Notably, less than half of participants (48.4%) were aware of CAM's harmful effects, and only 7.5% had received pre-service training in CAM practices (Table 2).

### Statistical differences in participants' awareness regarding CAM and its harmful effects by sociodemographic characteristics

Significant differences in CAM awareness were observed across several demographic variables. Age showed a significant association (p=0.007), with all physicians aged 35–40 years reporting CAM awareness. Female physicians demonstrated higher awareness (89.4%) compared to males (75.3%, p=0.025). Work experience significantly influenced both CAM awareness (p=0.045) and knowledge of harmful effects (p=0.005), with physicians having 10–19 years of experience showing the highest awareness (94.6%). Primary healthcare practitioners reported notably higher CAM awareness (90.1%) compared to other healthcare settings (p=0.012). Marital status was significantly associated with awareness of harmful effects (p=0.035), with married physicians showing greater awareness (Table 3).

**Table 1. Sociodemographic and occupational characteristics.**

| Characteristic | Missing | N (%) |
|---|---|---|
| Age (year) | 0 (0%) | |
| 25 to <30 | | 67 (42.1%) |
| 30 to <35 | | 38 (23.9%) |
| 35 to <40 | | 21 (13.2%) |
| ≥ 40 | | 33 (20.8%) |
| Gender | 0 (0%) | |
| Male | | 93 (58.5%) |
| Female | | 66 (41.5%) |
| Nationality | 20 (12.6%) | |
| Saudi | | 100 (71.9%) |
| Non-Saudi | | 39 (28.1%) |
| Marital Status | 0 (0%) | |
| Single | | 44 (27.7%) |
| Married | | 114 (71.7%) |
| Widowed | | 1 (0.6%) |
| Job level | 0 (0%) | |
| Intern | | 17 (10.7%) |
| Resident | | 70 (44.0%) |
| Specialist | | 45 (28.3%) |
| Consultant | | 23 (14.5%) |
| Others | | 4 (2.5%) |
| Work experience (years) | 0 (0%) | |
| < 10 | | 108 (67.9%) |
| 10 to <20 | | 37 (23.3%) |
| 20 to <30 | | 9 (5.7%) |
| ≥ 30 | | 5 (3.1%) |
| Current health facility of work | 0 (0%) | |
| Primary healthcare | | 81 (50.9%) |
| Secondary healthcare | | 32 (20.1%) |
| Tertiary healthcare | | 44 (27.7%) |
| Others | | 2 (1.3%) |
| Specialty | 1 (0.6%) | |
| Primary care and general practice | | 82 (51.9%) |
| Internal medicine or medical specialty | | 36 (22.8%) |
| Surgical specialty | | 15 (9.5%) |
| Paediatrics and emergency care | | 14 (8.9%) |
| Other specialities | | 11 (7.0%) |

## Pre-service training in CAM by sociodemographic characteristics

Pre-service training in CAM showed significant associations with several factors. Job level was significantly associated with training (p = 0.002), with specialists showing higher training rates (15.6%) compared to other positions. Work experience also showed significant differences (p = 0.026), with more experienced physicians (20–35 years) reporting higher training rates. Prior awareness of CAM (p = 0.009) and knowledge of its harmful effects (p < 0.001) were significantly associated with receiving pre-service training (Table 4).

**Table 2. Characteristics of participants' awareness and training of CAM.**

| Characteristic | Missing | N (%) |
|---|---|---|
| Ever heard about complementary medicine | 0 (0%) | 129 (81.1%) |
| If aware, the source of information about CAM* | 1 (0.8%) | |
| Books & lectures & webinars etc. | | 1 (0.8%) |
| Families & friends and relatives | | 55 (43.0%) |
| Health care providers | | 43 (33.6%) |
| Media (internet & television & radio and book) | | 90 (70.3%) |
| Patients using CAM | | 22 (17.2%) |
| Traditional healers | | 23 (18.0%) |
| From others | | 1 (0.8%) |
| Scientific journals | | 1 (0.8%) |
| Study | | 1 (0.8%) |
| University | | 1 (0.8%) |
| Complementary medicine treatment options you are aware of | 5 (3.1%) | |
| Acupuncture* | | 88 (57.1%) |
| Massage | | 112 (72.7%) |
| Meditation | | 72 (46.8%) |
| Yoga | | 83 (53.9%) |
| Deep-breathing exercises | | 75 (48.7%) |
| Hijama (Cupping) | | 113 (73.4%) |
| Traditional bone setting | | 35 (22.7%) |
| Medical herbalism | | 94 (61.0%) |
| Cautery | | 2 (1.3%) |
| Aware about the harmful effects of complementary medicine | 0 (0%) | |
| No | | 38 (23.9%) |
| Yes | | 77 (48.4%) |
| Do not know | | 44 (27.7%) |
| If yes, list the harmful effect(s) of complementary medicine* | 4 (5.2%) | |
| Abdominal pain | | 45 (61.6%) |
| Skin Discoloration | | 53 (72.6%) |
| Diarrhea | | 51 (69.9%) |
| Vomiting | | 45 (61.6%) |
| Others | | 30 (41.1%) |
| Ever received pre-service training in complementary medicine | 0 (0%) | |
| No | | 136 (85.5%) |
| Yes | | 12 (7.5%) |
| Non-available | | 11 (6.9%) |
| If yes, where do you get it? | 0 (0%) | |
| From a university or a college | | 9 (75.0%) |
| From a Health institution | | 3 (25.0%) |

*An asterisk indicates a multiple-choice item

## Participants' attitudes towards CAM

The highest positive attitudes among participants were towards the importance of conducting research about the efficacy and safety of complementary medicine with 82.4% agreeing or strongly agreeing. This was followed by 65.4% agreeing or

**Table 3. Statistical differences in participants' awareness regarding CAM and its harmful effects in terms of their sociodemographic characteristics.**

| Characteristic | Aware about CAM | | | Aware about harmful effects of CAM | | | |
|---|---|---|---|---|---|---|---|
| | No<br>N = 30 | Yes<br>N = 129 | p-value | No<br>N = 38 | Yes<br>N = 77 | Do not know<br>N = 44 | p-value |
| Age (year) | | | 0.007 | | | | 0.097 |
| 25 to < 30 | 20 (29.9%) | 47 (70.1%) | | 18 (26.9%) | 25 (37.3%) | 24 (35.8%) | |
| 30 to < 35 | 5 (13.2%) | 33 (86.8%) | | 10 (26.3%) | 17 (44.7%) | 11 (28.9%) | |
| 35 to < 40 | 0 (0.0%) | 21 (100.0%) | | 4 (19.0%) | 15 (71.4%) | 2 (9.5%) | |
| ≥ 40 | 5 (15.2%) | 28 (84.8%) | | 6 (18.2%) | 20 (60.6%) | 7 (21.2%) | |
| Gender | | | 0.025 | | | | 0.705 |
| Male | 23 (24.7%) | 70 (75.3%) | | 21 (22.6%) | 44 (47.3%) | 28 (30.1%) | |
| Female | 7 (10.6%) | 59 (89.4%) | | 17 (25.8%) | 33 (50.0%) | 16 (24.2%) | |
| Nationality | | | 0.088 | | | | 0.079 |
| Saudi | 23 (23.0%) | 77 (77.0%) | | 26 (26.0%) | 41 (41.0%) | 33 (33.0%) | |
| Non-Saudi | 4 (10.3%) | 35 (89.7%) | | 10 (25.6%) | 23 (59.0%) | 6 (15.4%) | |
| Marital Status | | | 0.400 | | | | 0.035 |
| Single | 11 (25.0%) | 33 (75.0%) | | 12 (27.3%) | 14 (31.8%) | 18 (40.9%) | |
| Married | 19 (16.7%) | 95 (83.3%) | | 26 (22.8%) | 62 (54.4%) | 26 (22.8%) | |
| Widowed | 0 (0.0%) | 1 (100.0%) | | 0 (0.0%) | 1 (100.0%) | 0 (0.0%) | |
| Job level | | | 0.293 | | | | 0.326 |
| Intern | 5 (29.4%) | 12 (70.6%) | | 4 (23.5%) | 6 (35.3%) | 7 (41.2%) | |
| Resident | 16 (22.9%) | 54 (77.1%) | | 21 (30.0%) | 29 (41.4%) | 20 (28.6%) | |
| Specialist | 5 (11.1%) | 40 (88.9%) | | 8 (17.8%) | 24 (53.3%) | 13 (28.9%) | |
| Consultant | 3 (13.0%) | 20 (87.0%) | | 5 (21.7%) | 14 (60.9%) | 4 (17.4%) | |
| Others | 1 (25.0%) | 3 (75.0%) | | 0 (0.0%) | 4 (100.0%) | 0 (0.0%) | |
| Work experience (years) | | | 0.045 | | | | 0.005 |
| < 10 | 24 (22.2%) | 84 (77.8%) | | 29 (26.9%) | 42 (38.9%) | 37 (34.3%) | |
| 10–19 | 2 (5.4%) | 35 (94.6%) | | 8 (21.6%) | 26 (70.3%) | 3 (8.1%) | |
| 20–29 | 3 (33.3%) | 6 (66.7%) | | 1 (11.1%) | 6 (66.7%) | 2 (22.2%) | |
| 30–35 | 1 (20.0%) | 4 (80.0%) | | 0 (0.0%) | 3 (60.0%) | 2 (40.0%) | |
| Current health facility of work | | | 0.012 | | | | 0.420 |
| Primary healthcare | 8 (9.9%) | 73 (90.1%) | | 20 (24.7%) | 42 (51.9%) | 19 (23.5%) | |
| Secondary healthcare | 8 (25.0%) | 24 (75.0%) | | 5 (15.6%) | 18 (56.3%) | 9 (28.1%) | |
| Tertiary healthcare | 14 (31.8%) | 30 (68.2%) | | 13 (29.5%) | 16 (36.4%) | 15 (34.1%) | |
| Others | 0 (0.0%) | 2 (100.0%) | | 0 (0.0%) | 1 (50.0%) | 1 (50.0%) | |
| Specialty | | | 0.054 | | | | 0.261 |
| Primary care and general practice | 9 (11.0%) | 73 (89.0%) | | 20 (24.4%) | 38 (46.3%) | 24 (29.3%) | |
| Internal medicine or medical specialty | 10 (27.8%) | 26 (72.2%) | | 5 (13.9%) | 19 (52.8%) | 12 (33.3%) | |
| Surgical specialty | 5 (33.3%) | 10 (66.7%) | | 5 (33.3%) | 6 (40.0%) | 4 (26.7%) | |
| Pediatrics and emergency care | 4 (28.6%) | 10 (71.4%) | | 7 (50.0%) | 5 (35.7%) | 2 (14.3%) | |
| Other specialties | 2 (18.2%) | 9 (81.8%) | | 1 (9.1%) | 8 (72.7%) | 2 (18.2%) | |

Data are expressed as n (%)

Fisher's exact test; Pearson's Chi-squared test

**Table 4.** Statistical differences in receiving pre-service training in terms of participants' sociodemographic characteristics and awareness levels.

| Characteristic | Ever received pre-service training | | | p-value |
|---|---|---|---|---|
| | **No**<br>**N = 136** | **Yes**<br>**N = 12** | **Non-available**<br>**N = 11** | |
| Age (year) | | | | 0.743 |
| 25 to <30 | 57 (85.1%) | 5 (7.5%) | 5 (7.5%) | |
| 30 to <35 | 32 (84.2%) | 2 (5.3%) | 4 (10.5%) | |
| 35 to <40 | 18 (85.7%) | 3 (14.3%) | 0 (0.0%) | |
| ≥ 40 | 29 (87.9%) | 2 (6.1%) | 2 (6.1%) | |
| Gender | | | | 0.050 |
| Male | 84 (90.3%) | 3 (3.2%) | 6 (6.5%) | |
| Female | 52 (78.8%) | 9 (13.6%) | 5 (7.6%) | |
| Nationality | | | | 0.917 |
| Saudi | 87 (87.0%) | 6 (6.0%) | 7 (7.0%) | |
| Non-Saudi | 34 (87.2%) | 3 (7.7%) | 2 (5.1%) | |
| Marital Status | | | | 0.887 |
| Single | 39 (88.6%) | 3 (6.8%) | 2 (4.5%) | |
| Married | 96 (84.2%) | 9 (7.9%) | 9 (7.9%) | |
| Widowed | 1 (100.0%) | 0 (0.0%) | 0 (0.0%) | |
| Job level | | | | **0.002** |
| Intern | 16 (94.1%) | 1 (5.9%) | 0 (0.0%) | |
| Resident | 62 (88.6%) | 0 (0.0%) | 8 (11.4%) | |
| Specialist | 37 (82.2%) | 7 (15.6%) | 1 (2.2%) | |
| Consultant | 19 (82.6%) | 2 (8.7%) | 2 (8.7%) | |
| Others | 2 (50.0%) | 2 (50.0%) | 0 (0.0%) | |
| Work experience (years) | | | | **0.026** |
| < 10 | 94 (87.0%) | 5 (4.6%) | 9 (8.3%) | |
| 10–19 | 33 (89.2%) | 4 (10.8%) | 0 (0.0%) | |
| 20–29 | 6 (66.7%) | 2 (22.2%) | 1 (11.1%) | |
| 30–35 | 3 (60.0%) | 1 (20.0%) | 1 (20.0%) | |
| Current health facility of work | | | | 0.922 |
| Primary healthcare | 67 (82.7%) | 8 (9.9%) | 6 (7.4%) | |
| Secondary healthcare | 28 (87.5%) | 2 (6.3%) | 2 (6.3%) | |
| Tertiary healthcare | 39 (88.6%) | 2 (4.5%) | 3 (6.8%) | |
| Others | 2 (100.0%) | 0 (0.0%) | 0 (0.0%) | |
| Specialty | | | | 0.244 |
| Primary care and general practice | 70 (85.4%) | 6 (7.3%) | 6 (7.3%) | |
| Internal medicine or medical specialty | 32 (88.9%) | 1 (2.8%) | 3 (8.3%) | |
| Surgical specialty | 14 (93.3%) | 0 (0.0%) | 1 (6.7%) | |
| Pediatrics and emergency care | 12 (85.7%) | 2 (14.3%) | 0 (0.0%) | |
| Other specialties | 7 (63.6%) | 3 (27.3%) | 1 (9.1%) | |
| Ever heard about complementary medicine | | | | **0.009** |
| No | 23 (76.7%) | 1 (3.3%) | 6 (20.0%) | |
| Yes | 113 (87.6%) | 11 (8.5%) | 5 (3.9%) | |

*(Continued)*

**Table 4.** (Continued)

| Characteristic | Ever received pre-service training | | | p-value |
|---|---|---|---|---|
| | No<br>N = 136 | Yes<br>N = 12 | Non-available<br>N = 11 | |
| Aware about the harmful effects of complementary medicine | | | | **<0.001** |
| No | 35 (92.1%) | 1 (2.6%) | 2 (5.3%) | |
| Yes | 68 (88.3%) | 9 (11.7%) | 0 (0.0%) | |
| Do not know | 33 (75.0%) | 2 (4.5%) | 9 (20.5%) | |

Data are expressed as n (%)

Fisher's exact test

strongly agreeing that medical practitioners should be more educated in the use of complementary medicine and 62.9% believing that providing both complementary and modern medicine for patients could increase patient satisfaction. Additionally, 55.4% of participants supported the provision of wellness centers using both complementary and modern medicine. In contrast, 17.6% of the participants strongly disagreed or disagreed about supporting the incorporation of complementary medicine in the medical curriculum"and 12.5% strongly disagreed or disagreed that medical practitioners should be more educated in the use of CAM (**Fig 1**).

Given that the responses to attitude items were homogenous (i.e., the responses were consistently reported on a Likert scale), we were able to assess the internal consistency of the attitudes' domain. Results of the reliability analysis showed a good level of internal consistency (Cronbach's alpha = 0.81). (Fig 2).

Only two factors showed significant differences in attitude scores. Job level was significantly associated with attitude scores (p = 0.026), with specialists showing higher median scores 24.0 (IQR 21.0, 25.0), compared to other positions. Prior awareness of CAM also showed significant differences (p = 0.016), with those aware of CAM having higher attitude scores with median 23.0 (IQR 20.0, 24.0) compared to those unaware (median = 20.0 (IQR 18.0, 23.0)).(Table 5).

Two factors showed significant associations with attitudes towards CAM. Age was significantly associated with CAM attitudes (p = 0.031), with younger physicians (30–35 years) showing more positive attitudes (68.4%) compared to older physicians (≥40 years, 33.3%). Job level also showed significant differences (p = 0.044), with specialists demonstrating more positive attitudes (68.9%) compared to consultants (34.8%) and other job levels (Table 6).

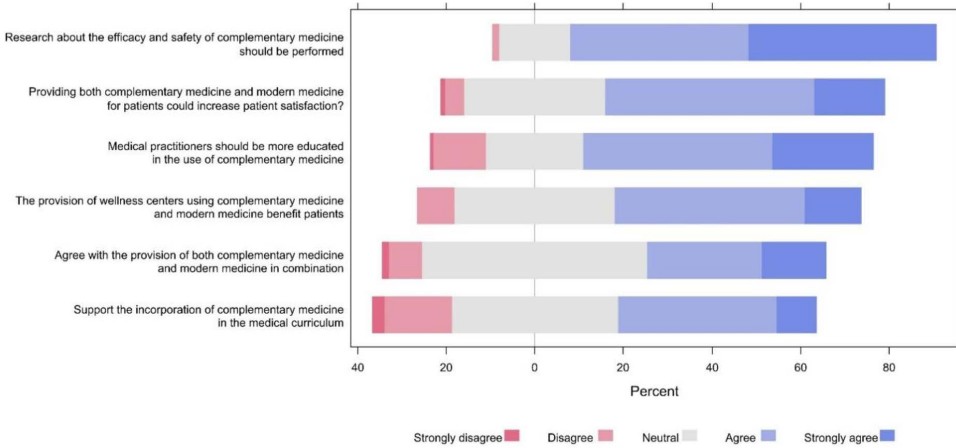

**Fig 1. The proportions of participants' responses to attitude items.**

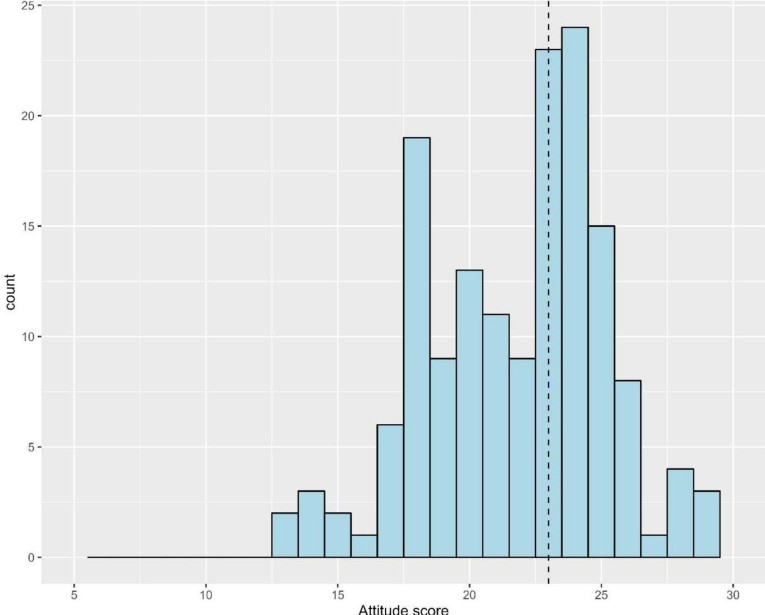

**Fig 2. A histogram depicting the frequency distribution of the attitudes score among physicians under study. The dashed line represents the median attitude score.**

### Participants' practice and recommendations regarding the use of CAM

Among the participants, 40.3% indicated that they might use complementary medicine (CAM) in the future, while 34.0% had no plans to use it, and 25.8% planned to use it. More than a half of physicians (61.0%) reported asking patients about their use of CAM, whereas 37.7% recommended CAM to their patients (**Fig 3**).

Regarding CAM practices and recommendations, 40.3% of participants indicated they might use CAM in the future, while 25.8% had definite plans to do so. A majority (61.0%) reported asking patients about CAM usage, though only 37.7% actively recommended CAM to patients. Among those who recommended CAM, massage (66.7%), deep-breathing exercises (55.0%), and Hijama (48.3%) were the most commonly recommended therapies. For personal use, 42.1% of participants reported using CAM in the past two years, with massage (57.6%) and deep-breathing exercises (50.0%) being the most common modalities. The main reasons for preferring CAM over modern medicine were acceptability (31.8%), accessibility (28.0%), and effectiveness (23.6%).(Table 7).

### Participants' Practice of Recommending CAM by Sociodemographic Characteristics and awareness levels

Only attitudes towards CAM showed a significant association with recommending CAM to patients (p < 0.001). Physicians with positive attitudes were more likely to recommend CAM (54.8%) compared to those with negative attitudes (18.7%). While not reaching statistical significance, there was a notable trend in current health facility of work (p = 0.063), with primary healthcare physicians showing higher recommendation rates (46.9%) compared to secondary (25.0%) and tertiary healthcare (29.5%) physicians (**Table 8**).

### Discussion

The current study was conducted to address the lack of comprehensive evidence on the KAP of physicians towards CAM in the Jazan region, a topic of growing concern on the global level. The findings revealed that although the proportion of

**Table 5. Statistical differences in attitude scores.**

| Characteristic | Median (IQR) | p-value |
|---|---|---|
| Age (year) | | 0.073 |
| 25–30 | 23.0 (19.0, 24.0) | |
| 30–35 | 23.0 (21.0, 25.0) | |
| 35–40 | 23.0 (21.0, 24.0) | |
| > 40 | 20.0 (18.0, 24.0) | |
| Gender | | 0.736 |
| Male | 23.0 (20.0, 24.0) | |
| Female | 23.0 (19.0, 24.0) | |
| Nationality | | 0.105 |
| Saudi | 23.0 (20.0, 25.0) | |
| Non-Saudi | 21.0 (18.0, 24.0) | |
| Marital Status | | 0.457 |
| Single | 22.0 (18.0, 24.0) | |
| Married | 23.0 (20.0, 24.0) | |
| Widowed | 19.0 (19.0, 19.0) | |
| Job level | | 0.026 |
| Intern | 23.0 (18.0, 25.0) | |
| Resident | 22.0 (19.0, 24.0) | |
| Specialist | 24.0 (21.0, 25.0) | |
| Consultant | 21.0 (18.0, 24.0) | |
| Others | 25.0 (23.5, 25.0) | |
| Work experience (years) | | 0.689 |
| < 10 | 23.0 (20.0, 24.0) | |
| 10–19 | 21.0 (18.0, 24.0) | |
| 20–29 | 20.0 (19.0, 24.0) | |
| 30–35 | 19.0 (18.0, 25.0) | |
| Current health facility of work | | 0.424 |
| Primary healthcare | 23.0 (20.0, 25.0) | |
| Secondary healthcare | 23.0 (18.0, 24.5) | |
| Tertiary healthcare | 21.0 (18.5, 24.0) | |
| Others | 21.5 (20.0, 23.0) | |
| Specialty | | 0.126 |
| Primary care and general practice | 23.0 (20.0, 25.0) | |
| Internal medicine or medical specialty | 21.0 (18.0, 24.0) | |
| Surgical specialty | 23.0 (18.0, 25.0) | |
| Pediatrics and emergency care | 23.0 (21.0, 24.0) | |
| Other specialties | 24.0 (21.0, 28.0) | |
| Ever heard about complementary medicine | | 0.016 |
| No | 20.0 (18.0, 23.0) | |
| Yes | 23.0 (20.0, 24.0) | |
| Aware about the harmful effects of complementary medicine | | 0.067 |
| No | 22.5 (20.0, 25.0) | |
| Yes | 23.0 (20.0, 24.0) | |
| Do not know | 21.5 (18.0, 24.0) | |

IQR: interquartile range

Kruskal-Wallis rank sum test; Wilcoxon rank sum test

**Table 6. Statistical differences in participants' attitudes towards CAM in terms of the sociodemographic characteristics.**

| Characteristic | Negative N=75 | Positive N=84 | p-value |
|---|---|---|---|
| Age (year) | | | **0.031** |
| 25 to <30 | 32 (47.8%) | 35 (52.2%) | |
| 30 to <35 | 12 (31.6%) | 26 (68.4%) | |
| 35 to <40 | 9 (42.9%) | 12 (57.1%) | |
| ≥ 40 | 22 (66.7%) | 11 (33.3%) | |
| Gender | | | 0.966 |
| Male | 44 (47.3%) | 49 (52.7%) | |
| Female | 31 (47.0%) | 35 (53.0%) | |
| Nationality | | | 0.155 |
| Saudi | 43 (43.0%) | 57 (57.0%) | |
| Non-Saudi | 22 (56.4%) | 17 (43.6%) | |
| Marital Status | | | 0.423 |
| Single | 23 (52.3%) | 21 (47.7%) | |
| Married | 51 (44.7%) | 63 (55.3%) | |
| Widowed | 1 (100.0%) | 0 (0.0%) | |
| Job level | | | **0.044** |
| Intern | 8 (47.1%) | 9 (52.9%) | |
| Resident | 37 (52.9%) | 33 (47.1%) | |
| Specialist | 14 (31.1%) | 31 (68.9%) | |
| Consultant | 15 (65.2%) | 8 (34.8%) | |
| Others | 1 (25.0%) | 3 (75.0%) | |
| Work experience (years) | | | 0.334 |
| < 10 | 46 (42.6%) | 62 (57.4%) | |
| 10–19 | 20 (54.1%) | 17 (45.9%) | |
| 20–29 | 6 (66.7%) | 3 (33.3%) | |
| 30–35 | 3 (60.0%) | 2 (40.0%) | |
| Current health facility of work | | | 0.722 |
| Primary healthcare | 35 (43.2%) | 46 (56.8%) | |
| Secondary healthcare | 15 (46.9%) | 17 (53.1%) | |
| Tertiary healthcare | 24 (54.5%) | 20 (45.5%) | |
| Others | 1 (50.0%) | 1 (50.0%) | |
| Specialty | | | 0.357 |
| Primary care and general practice | 38 (46.3%) | 44 (53.7%) | |
| Internal medicine or medical specialty | 22 (61.1%) | 14 (38.9%) | |
| Surgical specialty | 6 (40.0%) | 9 (60.0%) | |
| Pediatrics and emergency care | 5 (35.7%) | 9 (64.3%) | |
| Other specialties | 4 (36.4%) | 7 (63.6%) | |
| Ever heard about complementary medicine | | | 0.118 |
| No | 18 (60.0%) | 12 (40.0%) | |
| Yes | 57 (44.2%) | 72 (55.8%) | |
| Aware about the harmful effects of complementary medicine | | | 0.561 |
| No | 19 (50.0%) | 19 (50.0%) | |
| Yes | 33 (42.9%) | 44 (57.1%) | |
| Do not know | 23 (52.3%) | 21 (47.7%) | |

n (%)

Pearson's Chi-squared test; Fisher's exact test

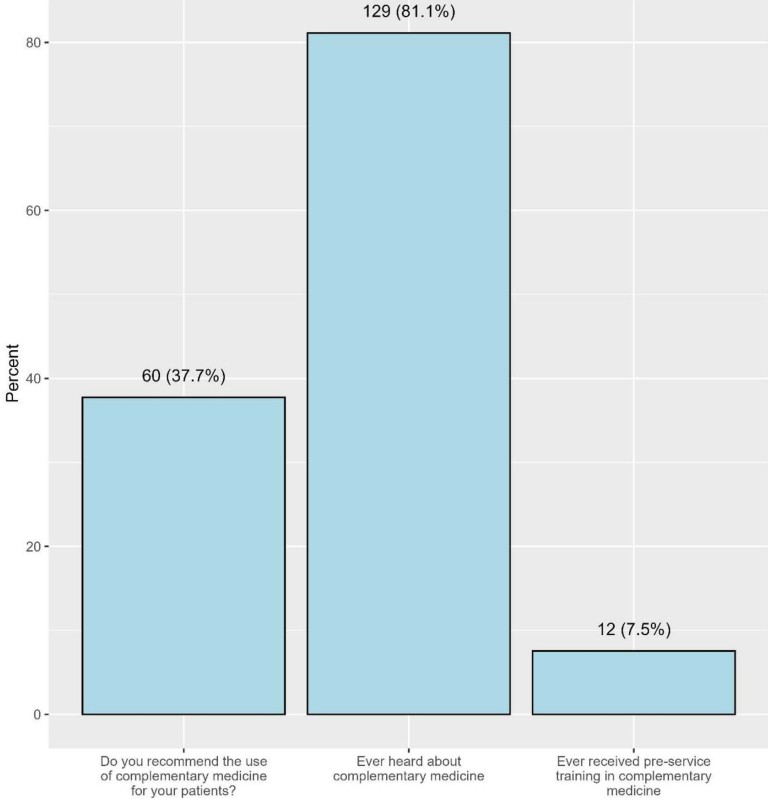

**Fig 3. The proportions of participants' awareness and practice and those who received preservice training about CAM.**

physicians who were aware of CAM was high (81.1%), only 7.5% of the participants had received formal pre-service training. This matter indicates a significant gap in education. Additionally, physicians' attitudes were generally positive since the median attitude score was 23 out of 30, reflecting the support for the use of CAM in healthcare practice. Nevertheless, concerning practical applications, about one-quarter of physicians planned to integrate CAM modalities, and more than one-third of them recommended CAM therapies. These results highlight the need for improved education and training on CAM to support the preparedness of healthcare providers to satisfy patients' needs and preferences [23].

The level of awareness in the current study towards CAM (81.1%) aligns with the results from other regions in Saudi Arabia despite some variations. For example, a survey carried out in Tabuk region showed that a vast majority of residents (95.8%) were aware about CAM [12], indicating a relatively higher level of awareness than in Jazan. Similarly, the majority of primary healthcare physicians in Riyadh (88.9%) had some knowledge of CAM, including cupping, honey and herbal medicine as the most recognized practices [24]. CAM awareness was also high in Qassim region among physicians (77.1%) [25]. On the other hand, a lower awareness level was apparent in Madinah, where almost three-quarters of physicians (72.9%) acknowledged the need to gain knowledge about CAM [13]This underscores a significant desire for education and training, although the level of awareness seems satisfactory. Accordingly, targeted initiatives that provide effective educational materials to bridge gaps and support the competency of healthcare professionals in CAM practices are needed. The educational need is paramount given the prominent lack of pre-service training [24].

The present study identified several factors associated with high awareness levels regarding CAM. These included age (higher awareness among those aged 35–40 years), gender (higher among females), work experience (higher for those with 10 to < 20 years of experience), and type of healthcare facility (higher awareness in primary healthcare settings).

**Table 7. Participants' responses to practice and recommendations to use CAM.**

| Characteristic | Missing | N (%) |
|---|---|---|
| Do you have plans to use complementary medicine in the future? | 0 (0%) | |
| No | | 54 (34.0%) |
| Maybe | | 64 (40.3%) |
| Yes | | 41 (25.8%) |
| Do you ask patients about complementary medicine usage? | 0 (0%) | 97 (61.0%) |
| Do you recommend the use of complementary medicine for your patients? | 0 (0%) | 60 (37.7%) |
| If yes, what types of complementary medicine do you typically recommend to your patients?* | 0 (0%) | |
| Acupuncture | | 11 (18.3%) |
| Massage | | 40 (66.7%) |
| Meditation | | 20 (33.3%) |
| Yoga | | 23 (38.3%) |
| Deep-breathing exercises | | 33 (55.0%) |
| Hijama | | 29 (48.3%) |
| Traditional bone setting | | 2 (3.3%) |
| Medical herbalism | | 25 (41.7%) |
| Reasons to prefer complementary medicine over modern medicine* | 2 (1.3%) | |
| Acceptability | | 50 (31.8%) |
| Effectiveness | | 37 (23.6%) |
| Accessibility | | 44 (28.0%) |
| Affordability | | 28 (17.8%) |
| Safety | | 2 (1.3%) |
| Have you used any complementary medicine in the last two years for yourself? | 0 (0%) | |
| No | | 82 (51.6%) |
| Maybe | | 10 (6.3%) |
| Yes | | 67 (42.1%) |
| If yes, what kinds of complementary medicine did you use?* | 1 (1.5%) | |
| Acupuncture | | 6 (9.1%) |
| Deep-breathing exercises | | 33 (50.0%) |
| Massage | | 38 (57.6%) |
| Yoga | | 12 (18.2%) |
| Meditation | | 18 (27.3%) |
| Herbal | | 5 (7.6%) |
| Hijama | | 17 (25.8%) |

*An asterisk indicates a multiple-choice item

The results are consistent with those reported in local studies [13,25]. For instance, physicians aged 30 or less showed lower awareness levels in Madinah [13], and experienced physicians reported higher familiarity with CAM, suggesting that professional experience is siganificant contributing factor to CAM awareness. Additionally, younger physicians and those with fewer years of experience in the United States have lower CAM awareness [26]. Furthermore, female physicians and those working in primary care were more likely to be aware abof CAM in Tabuk [12], which typically aligns with our findings. The same pattern of higher awareness among female doctors was observed in Canada [27]. Similar to our findings, specialists had higher awareness levels in Riyadh compared to general practitioners [10], indicating that professional exposure may play a role in increasing awareness regarding CAM.

**Table 8. Statistical differences in participants' practice regarding CAM in terms of their sociodemographic characteristics.**

| Characteristic | Recommending the use of CAM for patients | | p-value |
|---|---|---|---|
| | No N = 99 | Yes N = 60 | |
| Age (year) | | | 0.253 |
| 25 to < 30 | 43 (64.2%) | 24 (35.8%) | |
| 30 to < 35 | 19 (50.0%) | 19 (50.0%) | |
| 35 to < 40 | 13 (61.9%) | 8 (38.1%) | |
| ≥ 40 | 24 (72.7%) | 9 (27.3%) | |
| Gender | | | 0.764 |
| Male | 57 (61.3%) | 36 (38.7%) | |
| Female | 42 (63.6%) | 24 (36.4%) | |
| Nationality | | | 0.656 |
| Saudi | 60 (60.0%) | 40 (40.0%) | |
| Non-Saudi | 25 (64.1%) | 14 (35.9%) | |
| Marital Status | | | 0.667 |
| Single | 25 (56.8%) | 19 (43.2%) | |
| Married | 73 (64.0%) | 41 (36.0%) | |
| Widowed | 1 (100.0%) | 0 (0.0%) | |
| Job level | | | 0.123 |
| Intern | 11 (64.7%) | 6 (35.3%) | |
| Resident | 42 (60.0%) | 28 (40.0%) | |
| Specialist | 26 (57.8%) | 19 (42.2%) | |
| Consultant | 19 (82.6%) | 4 (17.4%) | |
| Others | 1 (25.0%) | 3 (75.0%) | |
| Work experience (years) | | | 0.703 |
| < 10 | 65 (60.2%) | 43 (39.8%) | |
| 10–19 | 23 (62.2%) | 14 (37.8%) | |
| 20–29 | 7 (77.8%) | 2 (22.2%) | |
| 30–35 | 4 (80.0%) | 1 (20.0%) | |
| Current health facility of work | | | 0.063 |
| Primary healthcare | 43 (53.1%) | 38 (46.9%) | |
| Secondary healthcare | 24 (75.0%) | 8 (25.0%) | |
| Tertiary healthcare | 31 (70.5%) | 13 (29.5%) | |
| Others | 1 (50.0%) | 1 (50.0%) | |
| Specialty | | | 0.132 |
| Primary care and general practice | 48 (58.5%) | 34 (41.5%) | |
| Internal medicine or medical specialty | 29 (80.6%) | 7 (19.4%) | |
| Surgical specialty | 8 (53.3%) | 7 (46.7%) | |
| Pediatrics and emergency care | 8 (57.1%) | 6 (42.9%) | |
| Other specialties | 6 (54.5%) | 5 (45.5%) | |
| Ever heard about complementary medicine | | | 0.071 |
| No | 23 (76.7%) | 7 (23.3%) | |
| Yes | 76 (58.9%) | 53 (41.1%) | |
| Aware about the harmful effects of complementary medicine | | | 0.330 |
| No | 20 (52.6%) | 18 (47.4%) | |

*(Continued)*

**Table 8.** (Continued)

| Characteristic | Recommending the use of CAM for patients | | p-value |
| --- | --- | --- | --- |
| | No<br>N = 99 | Yes<br>N = 60 | |
| Yes | 49 (63.6%) | 28 (36.4%) | |
| Do not know | 30 (68.2%) | 14 (31.8%) | |
| Attitudes towards CAM | | | <0.001 |
| Negative | 61 (81.3%) | 14 (18.7%) | |
| Positive | 38 (45.2%) | 46 (54.8%) | |

In the present study, results showed that the attitudes of physicians in the Jazan region were generally positive, with a median attitude score of 23, with the highest attitudes towards the importance of researchn CAM efficacy and safety and the need for improved education for healthcare professionals. These results are similar to those reported in Tabuk [12], where 74% of program residents had positive attitudes, and the participants highlighted the need for integrating conventional healthcare to improve patients' outcomes. On the international level, a cross-sectional survey in a medical centre in the United States showed that while several physicians were sceptical about CAM practices, they acknowledged the possible benefits of CAM for specific conditions [27], and this aligns with the positive attitudes shown in our study. Another study in the United Kingdom reported that physicians with CAM training and those who had previous experiences with CAM therapies were more likely to express positive attitudes and were open to integrating CAM into standard medical practice [28]. These observations suggest that while positive attitudes are significant among healthcare professionals globally and in Saudi Arabia, these attitudes are affected by specific factors, such as prior training, exposure and the need for further research and education on the efficacy and safety of CAM.

Our findings indicated that a relatively modest proportion of healthcare professionals in Jazan incorporated CAM into the clinical practice, with 25.8% and 37.7% planning to use CAM and recommending CAM therapies to their patients, respectively. In Tabuk, in a study involving 146 residents, almost a quarter of participants had previously used CAM in their practice, and more than half of them (52.1%) recommended its use as an adjunct to conventional medicine [12]. Additionally, 29% of physicians in Riyadh used CAM personally or professionally, although a few of them recommended CAM to patients [15]. In the United States, 38% of physicians recommended at least one type of CAM, reflecting a cautious but considerable level of integration of CAM into the clinical practice [27]. In Nigeria, 62.0% of physicians felt that herbal medicine remedies had a positive role in patients care [29]. Conversely, in the United Kingdom, only 19% of general practitioners recommended CAM modalities, possibly due to concerns about the lack of evidence-based suggestions about CAM efficacy [30]. In South Africa, fewer specialists (from 8% to 13%) felt that CAM provides a more holistic approach [31]. In general, there is a significant interest in CAM among physicians locally and internationally; however, the implementation of CAM is affected by the availability of evidence, patient demand, and personal beliefs.

The current study highlighted a significant gap between physicians recommending or endorsing CAM practices and their lack of formal training related to CAM. Such a critical issue is apparent when physicians encounter patients who are actively using CAM or seeking CAM-related advice. In our study, while only 7.5% of physicians reported receiving a formal training, 37.7% of had recommended CAM to their patients. These findings highlight a critical disconnect between public CAM utilization and physician preparedness. This gap between popular practice and professional preparation raises concerns about patient safety and healthcare quality. When physicians lack formal training in CAM, they may be ill-equipped to guide patients who are already using or seeking advice about these treatments, potentially missing opportunities for integrated care or failing to prevent harmful interactions with conventional treatments."

. Similar observations were reported by another study in Riyadh [24], with multiple physicians expressing interest in discussing CAM practices with patients while only 8% of them had attended CAM-related training. As such, there is an urgent need to integrate structured CAM education into medical training programs in order to ensure that physicians are well-equipped with knowledge to prescribe CAM safely and effectively.

The findings of the present study have shed light into the importance of CAM integration into the medical curriculum to enhance the attitudes of medical students in Saudi Arabia. A study conducted at Majmaa University showed that students who received formal education were more likely to be confident to understand and counsel patients on CAM practices [32]. In another study carried out at King Saud University and Majmaa University [33], students were interested in learning about CAM, and a significant proportion of them supported the inclusion of CAM content in the medical curriculum. Additionally, although a small proportion of students had taken a course in CAM (15%), the educated students had positive attitudes towards traditional modalities [33]. These results underline the significance of CAM incorporation into medical education to equip future physicians with the required skills and knowledge to address patients' interests in CAM. As a consequence, formal CAM education in medical schools might address the aforementioned gap by fostering evidence-based understanding of CAM therapies, including their benefits and potential risks. CAM integration in medical education would not only enhance physicians' competency, but also would promote safe and informed patient care.

From a practical perspective, CAM education and training can be improved via integrating CAM educational materials into undergraduate medical curricula and implementing continuing medical education programs and professional development opportunities. These scientific materials should focus on evidence-based CAM to assist in improving physicians' practices based on reliable therapies [33]. Such efforts would be ideally offered via structured training modules, practical exposure and providing consistent updates on novel evidence to ensure that physicians are well-prepared to use CAM effectively and safely.

Importantly, the current study did not explicitly investigate barriers to CAM integration into local clinical practice. However, since a significant proportion of physicians (65.4%) agreed that medical practitioners should be educated in CAM use, a gap in formal education seems to be a considerable barrier. Additionally, the lack of structured training was an apparent barrier, since only 7.5% of the participants had received pre-service training in CAM. The relatively small percentage of physicians planning to apply CAM in the clinical practice (25.8%) may also reflect a lack of confidence in knowledge regarding CAM modalities. These findings are in agreement with other studies which suggest that healthcare professionals are hesitant to integrate CAM due to limited evidence-based knowledge, lack of educational resources and lack of institutional support [16,34]. Future research should explore these barriers in details, potentially through mixed methods to assess the challenges faced by healthcare professionals in CAM integration into clinical practice.

Our study demonstrated significant variations in awareness and attitudes among physicians, which might be attributed to broader factors affecting different populations. Cultural beliefs ad societal norms are crucial determinants of CAM perceptions, particularly in Saudi Arabia. This is evident for distinct practices, like cupping therapy and prophetic medicine [24]. Furthermore, the lack of structured training and education can influence participants' attitudes and awareness, where limited exposure to CAM during medical education might decrease the level of confidence and understanding [35]. These observations might explain the reasons of having varying attitudes and levels of understanding of CAM, and these would help tailor effective solutions to increase physicians' attitudes.

The findings of the current study provide significant implications for practice and policy in the Jazan region and other areas with similar cultural patterns. The high levels of awareness, positive attitudes and the lack of formal pre-service training would all suggest the integration of CAM education into medical curricula and professional development programs to enhance CAM utilization in an effective and safe manner. Furthermore, tailored educational initiatives are required to address the specific needs of physicians' groups, particularly considering the observed demographic differences in awareness and attitudes. Future policies should bridge these gaps to enhance CAM utilization based on robust evidence. Indeed, the unique cultural and economic context of the Jazan region and similar regions should be considered when implementing changes in CAM education and integration. The widespread use of honey, Ruqyah and Hijama can be

exploited to support the acceptance of CAM education and training. The incorporation of these cultural beliefs and evidence-based CAM therapies into training programs can ultimately lead to an improved engagement. Economically, the integration of CAM practices into public healthcare offerings and subsidizing CAM services can reduce their costs and promote equitable access to CAM, especially in rural and unsupported areas.

However, the current study is not without limitations. The cross-sectional design may limit the ability to conclude the causal effects of demographic variables and different domains of physicians' KAP. Additionally, the self-reported nature of the survey might have limited the findings by introducing social desirability bias, where physicians might have overestimated their attitudes to align with the perceived expectations. Also, the convenience sampling approach allowed us to achieve a high response rate and include physicians from various healthcare settings, however, we acknowledge this as a limitation of our study. Although the sample under study was obtained from five regions in Jazan the study settings remain limited to a specific geographical area, which might restrict the generalizability of findings to the other areas inside and outside Saudi Arabia. Indeed, the specific cultural and healthcare context of the Jazan region might have contributed to unique patterns of cultural perceptions of CAM and access to training opportunities, which are different to other areas. Future studies might be warranted in other regions including, urban regions and diverse healthcare systems to assess whether similar patterns and barriers to CAM integration are observed.

Another limitation is the fact that the current study did not provide an in-depth analysis of the reasons behind physicians' attitudes and practices, and the survey did not consider the possible effects of cultural, religious and institutional factors that influence CAM integration into clinical practice. Additionally, the exclusion of non-English speaking physicians due to the language of the survey. This might have resulted in the underrepresentation of specific physicians within the Jazan region, especially those who usually communicate with such a language. Future studies should consider validated, bilingual instruments to optimize representativeness and inclusivity. Finally, since a small number of physicians had received formal training on CAM, the findings related to knowledge and practice should be interpreted with caution, as knowledge levels may not fully reflect the knowledge and competencies of other physician populations who received more comprehensive training. Future studies could address the above limitations by implementing a longitudinal design, incorporating mixed designs and recruiting a more diverse group of healthcare professionals.

## Conclusions

This study reveals critical insights into CAM integration within Saudi healthcare in the Jazan region. While most physicians are aware of CAM practices, only a small fraction have received formal pre-service training. This educational gap is particularly significant given that while the majority of physicians inquire about patients' CAM use, far fewer feel confident recommending these therapies. The study demonstrates a clear association between physicians' attitudes and their clinical practices: those with positive attitudes towards CAM are nearly three times more likely to recommend these therapies compared to those with negative attitudes.

These findings have important implications for medical education and healthcare policy in Saudi Arabia. The substantial gap between widespread awareness and limited formal training underscores the urgent need for structured CAM education in medical curricula and continuing professional development programs, particularly in primary healthcare settings where CAM integration shows the most promise. Future healthcare policies should prioritize evidence-based CAM training while developing frameworks that maintain high standards of patient care and acknowledge the role of complementary approaches in holistic healthcare delivery.

## Supporting information

**S1 File. dataset.**
(XLSX)

**S2 File. The survey instrument.**
(DOCX)

## Acknowledgments

The research team is grateful to all physicians who contributed to the study and to Dr. Mohamed S. Mahfouz for his help in sample size calculation and data analysis and to Mostafa A. Abdelmoaty from StatisMed for statistical analysis services for his help in data analysis and article draft refinement.

## Author contributions

**Conceptualization:** Sarah Salih, Tif A. Jawahi, Rafif H. Al Salem, Shahad A. Alhazmi, Atheer A. Buayti, Arwa H. Alammari, Hadeel M. Mashi, Layla A. Dobea, Mohammed A. Muaddi.

**Data curation:** Tif A. Jawahi, Rafif H. Al Salem, Shahad A. Alhazmi, Atheer A. Buayti, Arwa H. Alammari, Hadeel M. Mashi, Layla A. Dobea.

**Formal analysis:** Tif A. Jawahi.

**Methodology:** Sarah Salih, Mohammed A. Muaddi.

**Project administration:** Sarah Salih.

**Supervision:** Sarah Salih, Mohammed A. Muaddi.

**Writing – original draft:** Tif A. Jawahi, Rafif H. Al Salem, Shahad A. Alhazmi, Atheer A. Buayti, Arwa H. Alammari, Hadeel M. Mashi, Layla A. Dobea.

**Writing – review & editing:** Sarah Salih, Mohammed A. Muaddi.

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
