## [Decision Letter · Decision Letter 0]

14 Nov 2024

PONE-D-24-46538Bridging the Knowledge-Practice Gap: A Cross-sectional Survey Assessing Physician Knowledge, Attitude and Practice toward Complementary MedicinePLOS ONE

Dear Dr. Salih,

Thank you for submitting your manuscript to PLOS ONE. After careful consideration, we feel that it has merit but does not fully meet PLOS ONE’s publication criteria as it currently stands. Therefore, we invite you to submit a revised version of the manuscript that addresses the points raised during the review process.

We look forward to receiving your revised manuscript.

Kind regards,

Mohammed Hussain Abutaleb, PhD

Academic Editor

PLOS ONE

Journal Requirements:

 The authors gratefully acknowledge the funding of the Deanship of Graduate Studies and Scientific Research, Jazan University, Saudi Arabia, through Project Number: GSSRD-24 against publication costs of this article.  

The research team is grateful to the help and support of Dr. Mohamed S. Mahfouz in sample size calculation and data analysis. The team is also grateful to the help and support of Mostafa A. Abdelmoaty from StatisMed for statistical analysis services for his help in data analysis and article draft refinement. The authors gratefully acknowledge the funding of the Deanship of Graduate Studies and Scientific Research, Jazan University, Saudi Arabia, through Project Number: GSSRD-24

 The authors gratefully acknowledge the funding of the Deanship of Graduate Studies and Scientific Research, Jazan University, Saudi Arabia, through Project Number: GSSRD-24 against publication costs of this article.

4. Ethics statement appears in the Methods section of the manuscript AND at the end of the manuscript:

Your ethics statement should only appear in the Methods section of your manuscript. If your ethics statement is written in any section besides the Methods, please delete it from any other section. 

5. In the online submission form, you indicated that data are available from the corresponding author upon request. 

Reviewers' comments:

Reviewer's Responses to Questions

**Comments to the Author**

1. Is the manuscript technically sound, and do the data support the conclusions?

Reviewer #1: Yes

Reviewer #2: Yes

Reviewer #3: Partly

Reviewer #4: Yes

Reviewer #5: Yes

Reviewer #6: Yes

2. Has the statistical analysis been performed appropriately and rigorously? 

Reviewer #1: Yes

Reviewer #2: I Don't Know

Reviewer #3: Yes

Reviewer #4: I Don't Know

Reviewer #5: Yes

Reviewer #6: Yes

3. Have the authors made all data underlying the findings in their manuscript fully available?

Reviewer #1: Yes

Reviewer #2: No

Reviewer #3: Yes

Reviewer #4: Yes

Reviewer #5: Yes

Reviewer #6: Yes

4. Is the manuscript presented in an intelligible fashion and written in standard English?

Reviewer #1: Yes

Reviewer #2: No

Reviewer #3: No

Reviewer #4: Yes

Reviewer #5: Yes

Reviewer #6: Yes

5. Review Comments to the Author

Reviewer #1: The manuscript presents a significant and timely study that evaluates physicians' knowledge, attitudes, and practices regarding complementary and alternative medicine (CAM) in the Jazan region of Saudi Arabia. Given the rising global interest in CAM, it is crucial for healthcare providers to have a solid understanding and positive outlook on these practices to ensure effective communication with patients and promote safe usage.

Specific Comments:

1. Introduction:

1) The introduction lays a solid foundation regarding CAM and its global relevance. However, it would be beneficial for the authors to delve deeper into the specific reasons for conducting this study in the Jazan region. What particular characteristics or challenges in this area make the findings particularly valuable?

2. Methods:

1) The methods section is thorough and well-organized, detailing the study design, sampling methods, sample size calculation, and data analysis. Nonetheless, it would help to clarify the convenience sampling method used for physician recruitment and discuss any potential limitations this may introduce.

2)The authors mention that a validated survey was used for data collection. It would be useful to elaborate on the validation process of this survey and its reliability and validity specifically within the context of the Jazan region.

3. Results:

1) The results are clearly presented and well-structured. The use of both descriptive and inferential statistics effectively assesses the relationships between participants’ awareness, attitudes, and practices concerning CAM and their sociodemographic characteristics. However, a deeper exploration of the significance of these findings and their implications for practice and policy would enrich this section.

2) The manuscript states that the median attitude score was 23 out of 30 but does not provide a detailed breakdown of these scores across different demographic groups. Including this information would offer readers greater insight into the diversity of attitudes toward CAM among physicians.

4. Discussion:

1) The discussion is well-articulated and thoughtfully analyzes the findings. The authors effectively link the implications of their results to educational initiatives aimed at enhancing physicians’ competencies in CAM. That said, it would be advantageous to discuss potential barriers to incorporating CAM into clinical practice and propose specific strategies to address these obstacles.

2) The authors mention that the level of awareness of CAM aligns with findings from other regions in Saudi Arabia. However, a more detailed comparative analysis with those studies would strengthen the discussion, highlighting both similarities and differences in the results.

5. Limitations:

1) The authors briefly address the limitations of the study, such as the small sample size and the lack of an in-depth exploration of the reasons behind physicians' attitudes and practices. A more extensive discussion of how these limitations may affect the generalizability of the findings would be helpful.

6. Data Availability:

1) The authors indicate that all relevant data are included within the manuscript and supporting information files. However, it would be beneficial to clarify whether the survey instrument used is available for other researchers to utilize or replicate the study.

Minor Comments:

1. A thorough proofreading of the manuscript is recommended to catch any grammatical, punctuation, or spelling errors.

2. The authors should ensure consistency in formatting and citation style throughout the document.

Overall, this manuscript offers valuable insights into physicians’ knowledge, attitudes, and practices concerning CAM in the Jazan region of Saudi Arabia. With some revisions and clarifications, it holds great potential to contribute meaningfully to the field.

Reviewer #2: Thanks for your valuable investigation in terms on CAM. Some points mentioned below:

• The duration of sampling was 3 months which could be wider to earn more participant. Your involved 159 physicians, which might not be representative of the entire physician population in the Jazan region.

• Please give information about validity and reliability of your questionnaire

• The study notes that only 25.8% of physicians plan to use CAM in practice, but it does not explore the barriers to integration in detail. Identifying and addressing these barriers could provide more actionable insights.

• Consider whether the findings can be generalized to other regions or healthcare systems. The cultural and healthcare context of the Jazan region might limit the applicability of the results elsewhere.

• You said that only 25.8% of physicians plan to use CAM in practice, but it does not explore the barriers to integration in detail. Identifying and addressing these barriers could provide more actionable insights.

Reviewer #3: Comments have been uploaded and also check the manuscript below are the comments;

Dear Editor, the authors need to improve the following;

1. The title should be changed by adding “Alternative “ on the phrase complementary medicine i.e CAM

2. The abstract needs to be written according to the journal guidelines. The statistics reported in the results section need improvement, e.g.; female physicians were not mentioned anywhere.

3. Authors must provide references for most of the statements used in the manuscript.

4. Authors should not generalise their findings to health practitioners while they only focus on physicians, as it can lead to misleading conclusions.

5. A sample size calculation formula should be indicated, as it ensures the study is statistically valid and reliable.

6. The results section needs to be summarised. Not everything should be presented, especially tables.

7. The discussion section and the conclusion section have to be improved.

8. More comments and suggestions have been indicated in the manuscript as track changes.

Reviewer #4: The manuscript is good study exploring the KAP among Jazan physicians which is very important for easy integration of complementary medicine with current conventional medicine practices. Few comments are important to be addressed by the authors which are:

The title; I think didn't address the content of the study, I suggest to be " Knowledge, Attitude and Practice toward Complementary Medicine Among Physicians: A Cross Sectional Study in Jazan"

Introduction:

Please add a description of the current situation of complementary medicine regulation in Saudi Arabia, you can use this published manuscript with the Title "A new official national regulations for complementary medicine practices in Saudi Arabia". And with the title "The present state of complementary medicine regulation in Saudi Arabia."

Results:

Please do not repeat information which are presented in Tables and figures again in the Text.. you can just highlight the important findings,.

Discussion:

Please add a section regrading the attitude of Medical students in Saudi Arabia towards complementary medicine espicially after implementation of complementary medicine curriculum with their medical studies, you have examples from Majmaa University and Taibah University.

Add a paragraph regarding the importance of implementation of complementary medicine into medical curriculum.

Reviewer #5: I liked the topic raised in the paper. The study has very interesting data.

I believe that the problem of the population using different complementary medicines, some of which are prescribed by doctors, and the doctors themselves not having received adequate training for this, could be reinforced a little more (one paragraph).

The acronym KAP was used twice in the text; the meaning should have been mentioned as soon as it appeared for the first time, at the end of the Introduction.

I did not understand why non-English speakers were excluded. Why was this and whether it was possible to quantify how many there were? I believe that would be another relevant piece of information.

I did not understand when they mentioned positive attitude and how this was measured. I was confused about whether it was about doctors recommending CAM to patients, applying CAM in their practices, participating in studies and research on CAM, encouraging patients to use CAM...

Reviewer #6: This study assessed physicians’ awareness, attitudes, and practices regarding complementary and alternative medicine (CAM) in Jazan through a cross-sectional survey. Based on data collection and basic analysis, the study found that although physicians exhibit high awareness and positive attitudes towards CAM, they lack formal training and practical experience. In the future, it is necessary to provide them with more training and guidance.

Major:

1.Have you considered potential subjectivity in the survey responses? If so, please explain how it was addressed.

2.In the Methods section, you mention that “visiting physicians or non-English speakers were excluded.” Please provide evidence supporting the reasonableness of this exclusion criterion.

3.In the Methods section, you state that “a score higher than the median was considered indicative of a positive attitude toward CAM.” If all scores are low, this approach would still categorize half of the participants as having a positive attitude. Is this method truly rigorous?

4.In Table 4 of the Results section, while the p-value is <0.001, the difference between 3.3% and 8.5% is minimal, indicating that few physicians received pre-service training. Is this difference truly statistically significant?

5.Please analyze or explore the reasons that different populations may have varying attitudes and levels of understanding toward CAM.

6.In the Background and Introduction sections, provide conclusive evidence demonstrating that CAM is indeed effective in treating diseases.Relative reference such as PMID: 34896048 and PMID: 36939781 could be cited to introduce current application and research on CAM.

7.How do you propose improving CAM education and training?

8.Given the cultural and economic differences between this region and others, have you considered unique factors or alternative approaches to implementing changes?

9.Are you certain that South Korea has the highest rate of CAM usage in Asia, rather than China? This is somewhat surprising.

10.All tables currently display detailed data. Consider presenting only the most critical data in the tables, with additional data included in the supplementary materials.

Minor:

11.Please provide a legend for each graph.

12.Indicate how you conducted random sampling across the five regions.

13.Attach the complete questionnaire in the supplementary materials.

14.Many of the references are either outdated or from relatively lesser-known journals. Consider updating them as appropriate.

15.The study only includes two figures. Consider adding more visual representations of the data.

16.In the Conclusion section on the second page, there are two consecutive commas—please correct this.

6. PLOS authors have the option to publish the peer review history of their article (what does this mean? ). If published, this will include your full peer review and any attached files.

**Do you want your identity to be public for this peer review?** For information about this choice, including consent withdrawal, please see our Privacy Policy .

Reviewer #1: No

Reviewer #2: No

Reviewer #3: **Yes: ** Ivan Kahwa

Reviewer #4: No

Reviewer #5: **Yes: ** Kassia Martins Fernandes Pereira

Reviewer #6: **Yes: ** Chen Ling

---

## [Author Response · Author response to Decision Letter 1]

28 Dec 2024

Dear respected editor and reviewers,

Thank you very much for allowing us to revise our manuscript. We are grateful to the Editor and reviewers for evaluating our manuscript and providing constructive comments to improve its quality. We have revised the manuscript to address all the points raised during the review process. Kindly find our responses to the outstanding reviewers in the file labeled response to reviewers (attached).

Journal Comments

Journal Requirements Responses

The manuscript was amended according to the PLOS ONE style except for the results and discussion section. The results section is extensive in this manuscript and the authors felt that combining it with the discussion would make things confusing for readers. However, we are willing to separate these two sections if the journal requires so. We appreciate your understanding.

The authors gratefully acknowledge the funding of the Deanship of Graduate Studies and Scientific Research, Jazan University, Saudi Arabia, through Project Number: GSSRD-24 against publication costs of this article.

Financial disclosure was amended as follow: The authors gratefully acknowledge the funding of the Deanship of Graduate Studies and Scientific Research, Jazan University, Saudi Arabia, through Project Number: GSSRD-24. The funders had no role in study design, data collection and analysis, decision to publish, or preparation of the manuscript and support was for publication costs only.

The research team is grateful to the help and support of Dr. Mohamed S. Mahfouz in sample size calculation and data analysis. The team is also grateful to the help and support of Mostafa A. Abdelmoaty from StatisMed for statistical analysis services for his help in data analysis and article draft refinement. The authors gratefully acknowledge the funding of the Deanship of Graduate Studies and Scientific Research, Jazan University, Saudi Arabia, through Project Number: GSSRD-24

The authors gratefully acknowledge the funding of the Deanship of Graduate Studies and Scientific Research, Jazan University, Saudi Arabia, through Project Number: GSSRD-24 against publication costs of this article.

We removed any funding-related text from the manuscript and updated the funding statement as mentioned above.

4. Ethics statement appears in the Methods section of the manuscript AND at the end of the manuscript:

Your ethics statement should only appear in the Methods section of your manuscript. If your ethics statement is written in any section besides the Methods, please delete it from any other section.

Ethics statement was removed except from the Methods section.

5. In the online submission form, you indicated that data are available from the corresponding author upon request.

Data described in the manuscript is provided freely as supplementary files and the statement was amended to:

The datasets used and/or analyzed during the current study are available as supplementary information under the name S1_dataset. The survey instrument used is available as supplementary file S2_ Instrument.

Reviewer 1

The manuscript presents a significant and timely study that evaluates physicians' knowledge, attitudes, and practices regarding complementary and alternative medicine (CAM) in the Jazan region of Saudi Arabia. Given the rising global interest in CAM, it is crucial for healthcare providers to have a solid understanding and positive outlook on these practices to ensure effective communication with patients and promote safe usage.

The introduction lays a solid foundation regarding CAM and its global relevance. However, it would be beneficial for the authors to delve deeper into the specific reasons for conducting this study in the Jazan region. What particular characteristics or challenges in this area make the findings particularly valuable?

Thank you for your comment. We added the following sequence to address the comment:

“The Jazan region has unique geographic, cultural and healthcare landscape, since the population in this area is predominantly rural, with strong adherence to traditional and cultural practices [11, 12]. Additionally, there is evidence that the access to advanced healthcare services is limited in some local areas [13], which makes the choice of CAM a popular approach.”

2. Methods:

1) The methods section is thorough and well-organized, detailing the study design, sampling methods, sample size calculation, and data analysis. Nonetheless, it would help to clarify the convenience sampling method used for physician recruitment and discuss any potential limitations this may introduce. Thank you. We added the following sequence in “Sampling technique and sample size” subsection under the methods section:

“Such a sampling technique was used to allow efficient recruitment and inclusion of diverse healthcare settings. Efforts were made to reduce the potential selection bias (physicians who were more available or willing to participate may differ in their awareness) and enhance the representativeness of participants. This was done by recruiting participants from a variety of governorates and healthcare settings.”

2)The authors mention that a validated survey was used for data collection. It would be useful to elaborate on the validation process of this survey and its reliability and validity specifically within the context of the Jazan region.

We already provided the following statement: “Given that the responses to attitude items were homogenous (i.e. the responses were consistently reported on a Likert scale), we were able to assess the internal consistency of the attitudes’ domain. Results of the reliability analysis showed a good level of internal consistency (Cronbach’s alpha = 0.812).”

The reliability of other domains was not calculated because they included responses with heterogenous responses (not consistent Likert scales) and multiple choice items.

The following paragraph was added:

"The survey instrument underwent a rigorous validation process specifically adapted for the Jazan context. Initially, the questionnaire was developed in English based on comprehensive literature review. Content validity was established through review by a panel of local experts (n=5) including CAM specialists, research methodologists, and practicing physicians from Jazan region who assessed item relevance, clarity, and cultural appropriateness. The Content Validity Index (CVI) was calculated, with items achieving a minimum CVI of 0.80 being retained.

3. Results:

1) The results are clearly presented and well-structured. The use of both descriptive and inferential statistics effectively assesses the relationships between participants’ awareness, attitudes, and practices concerning CAM and their sociodemographic characteristics. However, a deeper exploration of the significance of these findings and their implications for practice and policy would enrich this section.

Thank you for the suggestion. We acknowledge the importance of discussing the implications of the findings in greater depth. To maintain the structure of the manuscript, we have added a sequence to the Discussion section, as the results section is intended solely to present the study's findings without interpretation or implications. The sequence is:

“The findings of the current study provide significant implications for practice and policy in the Jazan region and other areas with similar cultural patterns. The high levels of awareness, positive attitudes and the lack of formal pre-service training would all suggest the integration of CAM education into medical curricula and professional development programs to enhance CAM utilization in an effective and safe manner. Furthermore, tailored educational initiatives are required to address the specific needs of physicians’ groups, particularly considering the observed demographic differences in awareness and attitudes. Future policies should bridge these gaps to enhance CAM utilization based on robust evidence.”

2) The manuscript states that the median attitude score was 23 out of 30 but does not provide a detailed breakdown of these scores across different demographic groups. Including this information would offer readers greater insight into the diversity of attitudes toward CAM among physicians. Thank you for your comment. We added table 5 providing details about the breakdown of attitudes scores as suggested.

4. Discussion:

1) The discussion is well-articulated and thoughtfully analyzes the findings. The authors effectively link the implications of their results to educational initiatives aimed at enhancing physicians’ competencies in CAM. That said, it would be advantageous to discuss potential barriers to incorporating CAM into clinical practice and propose specific strategies to address these obstacles.

We added the following paragraph to the discussion section to discuss potential barriers, with future recommendations:

“Importantly, the current study did not explicitly investigate barriers to CAM integration into local clinical practice. However, since a significant proportion of physicians (65.4%) agreed that medical practitioners should be educated in CAM use, a gap in formal education seems to be a considerable barrier. Additionally, the lack of structured training was an apparent barrier, since only 7.5% of the participants had received pre-service training in CAM. The relatively small percentage of physicians planning to apply CAM in the clinical practice (25.8%) may also reflect a lack of confidence in knowledge regarding CAM modalities. These findings are in agreement with other studies which suggest that healthcare professionals are hesitant to integrate CAM due to limited evidence-based knowledge, lack of educational resources and lack of institutional support [18, 19]. Future research should explore these barriers in details, potentially through mixed methods to assess the challenges faced by healthcare professionals in CAM integration into clinical practice.”

2) The authors mention that the level of awareness of CAM aligns with findings from other regions in Saudi Arabia. However, a more detailed comparative analysis with those studies would strengthen the discussion, highlighting both similarities and differences in the results.

Thank you for your comment. We added the following paragraph to address the comparative analysis with other Saudi studies:

The level of awareness in the current study towards CAM (81.1%) aligns with the results from other regions in Saudi Arabia despite some variations. For example, a survey carried out in Tabuk region showed that a vast majority of residents (95.8%) were aware about CAM [7], indicating a relatively higher level of awareness than in Jazan. Similarly, the majority of primary healthcare physicians in Riyadh (88.9%) had some knowledge of CAM, including cupping, honey and herbal medicine as the most recognized practices [18]. CAM awareness was also high in Qassim region among physicians (77.1%) [19]. On the other hand, a lower awareness level was apparent in Madinah, where almost three-quarters of physicians (72.9%) acknowledged the need to gain knowledge about CAM [8]This underscores a significant desire for education and training, although the level of awareness seems satisfactory. Accordingly, targeted initiatives that provide effective educational materials to bridge gaps and support the competency of healthcare professionals in CAM practices are needed. The educational need is paramount given the prominent lack of pre-service training [18].

5. Limitations:

1) The authors briefly address the limitations of the study, such as the small sample size and the lack of an in-depth exploration of the reasons behind physicians' attitudes and practices. A more extensive discussion of how these limitations may affect the generalizability of the findings would be helpful. We provided more details about the generalizability of findings in the following sequence:

“Although the sample under study was obtained from five regions in Jazan, the study settings remain limited to a specific geographical area, which might restrict the generalizability of findings to the other areas inside and outside Saudi Arabia. Indeed, the specific cultural and healthcare context of the Jazan region might have contributed to unique patterns of cultural perceptions of CAM and access to training opportunities, which are different to other areas. Future studies might be warranted in other regions including, urban regions and diverse healthcare systems to assess whether similar patterns and barriers to CAM integration are observed.”

6. Data Availability:

1) The authors indicate that all relevant data are included within the manuscript and supporting information files. However, it would be beneficial to clarify whether the survey instrument used is available for other researchers to utilize or replicate the study. We added the used survey in the supplementary material (S2_ Instrument).

Minor Comments:

1. A thorough proofreading of the manuscript is recommended to catch any grammatical, punctuation, or spelling errors The manuscript has been edited by a professional proofreader.

2. The authors should ensure consistency in formatting and citation style throughout the document.

The formatting and citation styles have been confirmed for consistency.

Overall, this manuscript offers valuable insights into physicians’ knowledge, attitudes, and practices concerning CAM in the Jazan region of Saudi Arabia. With some revisions and clarifications, it holds great potential to contribute meaningfully to the field.

Reviewer 2

Thanks for your valuable investigation in terms on CAM. Some points mentioned below:

Reviewer comments Responses

The duration of sampling was 3 months which could be wider to earn more participant. Your involved 159 physicians, which might not be representative of the entire physician population in the Jazan region. Thank you for your comment. The sampling duration of three months was sufficient to reach the target sample size of 159 p

---

## [Decision Letter · Decision Letter 1]

26 Mar 2025

Bridging the Knowledge-Practice Gap: A Cross-sectional Survey Assessing Physician Knowledge, Attitude and Practice toward Complementary and Alternative Medicine

PONE-D-24-46538R1

Dear Dr. Salih,

We’re pleased to inform you that your manuscript has been judged scientifically suitable for publication and will be formally accepted for publication once it meets all outstanding technical requirements.

Kind regards,

Mohammed Abutaleb, PhD

Academic Editor

PLOS ONE

Additional Editor Comments (optional):

Reviewers' comments:

Reviewer's Responses to Questions

**Comments to the Author**

1. If the authors have adequately addressed your comments raised in a previous round of review and you feel that this manuscript is now acceptable for publication, you may indicate that here to bypass the “Comments to the Author” section, enter your conflict of interest statement in the “Confidential to Editor” section, and submit your "Accept" recommendation.

Reviewer #2: All comments have been addressed

Reviewer #7: All comments have been addressed

Reviewer #8: All comments have been addressed

2. Is the manuscript technically sound, and do the data support the conclusions?

Reviewer #2: Yes

Reviewer #7: Yes

Reviewer #8: Yes

3. Has the statistical analysis been performed appropriately and rigorously? 

Reviewer #2: I Don't Know

Reviewer #7: Yes

Reviewer #8: Yes

4. Have the authors made all data underlying the findings in their manuscript fully available?

Reviewer #2: Yes

Reviewer #7: Yes

Reviewer #8: Yes

5. Is the manuscript presented in an intelligible fashion and written in standard English?

Reviewer #2: Yes

Reviewer #7: Yes

Reviewer #8: Yes

6. Review Comments to the Author

Reviewer #2: Thank you so much you worked on a god topic and I hope these works will be continued in the future.

All comments addressed very well and there is no additional comments.

Reviewer #7: (No Response)

Reviewer #8: All the comments are addressed satisfactorily and the manuscript can now be accepted in its present form.

7. PLOS authors have the option to publish the peer review history of their article (what does this mean? ). If published, this will include your full peer review and any attached files.

**Do you want your identity to be public for this peer review?** For information about this choice, including consent withdrawal, please see our Privacy Policy .

Reviewer #2: **Yes: ** JAVAD NADALI

Reviewer #7: No

Reviewer #8: **Yes: ** Waquar Ahsan

---

## [Editor Report · Acceptance letter]

PONE-D-24-46538R1

PLOS ONE

Dear Dr. Salih,

I'm pleased to inform you that your manuscript has been deemed suitable for publication in PLOS ONE. Congratulations! Your manuscript is now being handed over to our production team.

Kind regards,

on behalf of

Dr. Mohammed Abutaleb

Academic Editor

PLOS ONE